# Harnessing Nuclear Energy to Gold Nanoparticles for the Concurrent Chemoradiotherapy of Glioblastoma

**DOI:** 10.3390/nano13212821

**Published:** 2023-10-24

**Authors:** Jui-Ping Li, Yu-Cheng Kuo, Wei-Neng Liao, Ya-Ting Yang, Sih-Yu Chen, Yu-Ting Chien, Kuo-Hung Wu, Mei-Ya Wang, Fong-In Chou, Mo-Hsiung Yang, Dueng-Yuan Hueng, Chung-Shi Yang, Jen-Kun Chen

**Affiliations:** 1Institute of Biomedical Engineering and Nanomedicine, National Health Research Institutes, Miaoli 35053, Taiwan; piny72@nhri.edu.tw (J.-P.L.); 970730@nhri.edu.tw (W.-N.L.); fructose0129@hotmail.com (Y.-T.Y.); sihyu2011@gmail.com (S.-Y.C.); ken800322@gmail.com (Y.-T.C.); cyang@nhri.edu.tw (C.-S.Y.); 2Department of Radiation Oncology, China Medical University Hospital, Taichung 40447, Taiwan; shapico22@gmail.com; 3School of Medicine, China Medical University, Taichung 40402, Taiwan; 4Nuclear Science and Technology Development Center, National Tsing Hua University, Hsinchu 30013, Taiwan; khwu@mx.nthu.edu.tw (K.-H.W.); meywang@mx.nthu.edu.tw (M.-Y.W.); fichou@mx.nthu.edu.tw (F.-I.C.); 5Department of Biomedical Engineering and Environmental Sciences, National Tsing Hua University, Hsinchu 30013, Taiwan; mhyang@mx.nthu.edu.tw; 6School of Medicine, National Defense Medical Center, Taipei 11490, Taiwan; hondy2195@yahoo.com.tw; 7Biotechnology Center, National Chung Hsing University, Taichung 40227, Taiwan; 8Graduate Institute of Life Sciences, National Defense Medical Center, Taipei 11490, Taiwan

**Keywords:** radioactive gold nanoparticles, one-pot/one-step reaction, nuclear energy, glioblastoma, ^198^Au, temozolomide, concurrent chemoradiotherapy

## Abstract

Nuclear fission reactions can release massive amounts of energy accompanied by neutrons and γ photons, which create a mixed radiation field and enable a series of reactions in nuclear reactors. This study demonstrates a one-pot/one-step approach to synthesizing radioactive gold nanoparticles (RGNP) without using radioactive precursors and reducing agents. Trivalent gold ions are reduced into gold nanoparticles (8.6–146 nm), and a particular portion of ^197^Au atoms is simultaneously converted to ^198^Au atoms, rendering the nanoparticles radioactive. We suggest that harnessing nuclear energy to gold nanoparticles is feasible in the interests of advancing nanotechnology for cancer therapy. A combination of RGNP applied through convection-enhanced delivery (CED) and temozolomide (TMZ) through oral administration demonstrates the synergistic effect in treating glioblastoma-bearing mice. The mean survival for RGNP/TMZ treatment was 68.9 ± 9.7 days compared to that for standalone RGNP (38.4 ± 2.2 days) or TMZ (42.8 ± 2.5 days) therapies. Based on the verification of bioluminescence images, positron emission tomography, and immunohistochemistry inspection, the combination treatment can inhibit the proliferation of glioblastoma, highlighting the niche of concurrent chemoradiotherapy (CCRT) attributed to RGNP and TMZ.

## 1. Introduction

Modern nanotechnology started with the publication of the first scientific article discussing the colloidal properties of gold nanoparticles (GNP) [1,2], written by Michael Faraday in 1857 [3]. The GNP has been extensively studied and developed for multiple applications, such as a research tool for life science, a key component of diagnostic kits, a contrast agent of X-ray imaging, and a drug carrier of nanomedicine. Since the 1970s, immunogold, consisting of GNP conjugated with antibodies, has been designed to recognize the location of target proteins on the biological specimens since gold is an electron-dense element providing outstanding contrast in the use of transmission electron microscopy (TEM) [4,5]. The GNP-based pregnancy test to detect urinary human chorionic gonadotropin (hCG) went on the market in the 1980s [6,7]. Owing to excellent optical and electrical properties, GNP and its composites are frequently used to develop sensing devices [8,9,10] and diagnostic kits [11,12].

GNP has played a remarkable role in nanomedicine, especially in drug delivery systems for cancer therapy. GNP with optimal particle size has demonstrated superior tumor penetration and retention performance in tumors [13]. Chemotherapeutics-conjugated GNP targeting of tumors is attributed to their enhanced permeation and retention (EPR) effects, leading to more internalized drugs in tumor cells than molecular drugs [13,14,15]. Gibson et al. and Hwu et al. have reported the conjugation of Paclitaxel to GNP with a mean diameter of 2 nm and 14.6 nm, respectively, estimating an average of 77 or 201 paclitaxel molecules on gold nanoparticles [16,17]. Similarly, Brown et al. demonstrated an improvement in oxaliplatin delivery through the conjugation of 280 oxaliplatin molecules on gold nanoparticles with mean diameters of 31 nm [18]. Doxorubicin should be a remarkable chemotherapeutic when bound onto GNP, increasing the drug molecules trafficked into the tumor region [19,20,21]. 

Regarding therapeutic efficacy, drug-loaded nanocarriers are limited by the heterogeneity of the EPR effect within and between different tumors [22]. High-EPR tumors are usually observed in subcutaneous tumors in xenografts, owing to rapid tumor growth, which rarely happens in humans. Wilhelm et al. indicated that only 0.7% (median) of injected doses (ID) of nanocarriers reached the target tumors in a meta-analysis of preclinical data [23]. Further, van Vlerken et al. presented a preclinical study showing a delivery efficiency of 0.6% ID for paclitaxel-loaded nanocarriers compared to 0.2% ID for free paclitaxel [24]. Therefore, overcoming multiple physiological barriers to nanoparticle drug delivery remains challenging. 

There have been eight clinical trials of therapeutic gold nanostructures, but just three of them are GNP-based drug carriers for anticancer. The research group led by Voliani has indicated that clinical translation of metal-based nanoparticles, especially for those with a size larger than 20 nm, was prevented because of its persistence in organs after medical action [25]. A promising approach to reducing particle size to ultrasmall gold nanoparticles (<5 nm) is highlighted due to the excretion through the renal/urinary pathway [26,27]. Furthermore, an increase in the specific surface area on ultrasmall nanoparticles can additionally overcome the limitations of drug loading content compared to conventional larger nanoparticles (>20 nm) [28], eliminating the use of nanoparticles and probable side effects caused by them.

Instead of small molecular drugs, conjugating GNP with high-potency therapeutics, such as tumor necrosis factor-alpha (TNF-α), shows a niche superior to conjugating GNP with chemotherapeutics [29,30,31]. The TNF-α presents high potency with minimal drug-loading content since polyvalent interactions exist between targets and TNF-α molecules. Furthermore, radioactive isotopes can feasibly conjugate with GNP to generate radioactive gold nanoparticles (RGNP) for diagnostic and therapeutic purposes. The ^111^In, ^124^I, and ^125^I were used to prepare RGNP for tracking them in vivo, suggesting ^125^I-labeled GNP for quantifying GNP in real time [32], ^111^In-labeled GNP for SPECT imaging [33], and ^124^I-labeled GNP for PET imaging [34]. Also, ^198^Au-GNP has been developed for cancer therapy. Kannan and Katti’s group presented the idea of using radioactive aqueous H^198^AuCl_4_ as a precursor to synthesize gum arabic glycoprotein-functionalized gold nanoparticles for treating prostate cancer [35,36,37]. Zhou et al. showed the synthesis of glutathione-coated radioactive gold nanoparticles through a one-step reaction using ^198^Au as the precursor and glutathione as a reducing agent [38]. Chen et al. demonstrated a method of GNP prepared using the Turkevich method followed by thermal neutron irradiation to produce ^198^Au-GNP [39].

To produce GNP rapidly, Wang et al. attempted to use a radiation-induced chemical reaction to synthesize naked gold nanoparticles, in which synchrotron white X-ray photons (1.0 × 10^12^ photons/s) centered at around 12.5 keV lead to spherical GNP with a size of 20 ± 5 nm [40]. A mixed radiation field in a nuclear reactor provides γ photons and neutrons. The γ photons play a role as the white X-ray photons, and then thermal neutrons can transform ^197^Au into ^198^Au through neutron capture reaction without using radioactive precursors and reducing agents. Therefore, we propose that nuclear energy might enable RGNP production through a one-pot/one-step reaction in a nuclear reactor. From the viewpoint of medical application, the ^198^Au-GNP takes advantage of dual functions since simultaneous emission of γ photons (412 keV) and β particles (E_βmax_: 0.96 MeV) with a half-life of 2.69 days [41]. The γ photon scintigraphy of ^198^Au-GNP in living animals using single-photon-emission-computed tomography (SPECT) has been demonstrated in previous work [39]. The β particles emitting from ^198^Au-GNP can provide 0.61 MeV of maximal kinetic energy, 58% higher than those from ^131^I, thereby indicating better penetration in soft tissues of locoregional radiotherapy.

Glioblastoma multiform (GBM) is well-recognized as the most aggressive and lethal malignant tumor, presenting 14.6 months of median survival and less than 5% of patients’ 3-year overall survival rate [42]. So far, treating GBM remains highly challenging. Maximal surgical resection could be the first-line treatment if the tumor is operable. Radiotherapy with adjuvant chemotherapy (or concurrent chemoradiotherapy) should be a standard procedure to prevent tumor recurrence. Unfortunately, recurrent GBM could eventually take patients’ lives since the tumors are finally inoperable and can not receive more doses from external beam radiotherapy. The GNP could be an ideal radiation carrier because of the previous success of radioactive gold seeds for brachytherapy and fiducial markers for prostate image-guided radiation therapy (IGRT). Furthermore, GNP can be formulated into injectable solutions, which could be used for treating recurrent/inoperable glioblastoma through convection-enhanced delivery (CED).

Despite the fact that the blood–brain barrier (BBB) is always a critical challenge hindering drug delivery into the brain, not only for small-molecule drugs but also for nanomedicine [22,43]. Rosenblum et al. suggested that the local administration of therapeutics directly into the diseased compartment could be a reasonable strategy [22]. The CED takes advantage of simultaneous control of local–regional delivery, duration of drug release, and diffusion. The CED has been successful in delivering platins [44,45], irinotecan-loaded liposome [46], and doxorubicin/epirubicin-loaded virus-like nanoparticles [47,48]. The ClinicalTrial.gov website has collected 35 registered trials (the last accessed day: 20 June 2023) of CED, and most of them are applied to treat glioma. Due to chemotherapy and radiotherapy being the most frequently used adjuvant therapy for GBM patients after surgery, we anticipate integrating temozolomide (TMZ) through oral administration and ^198^Au-GNP through CED to treat intracranial glioblastoma xenografts, which highlights the potential of RGNP prepared through a one-pot/one-step nuclear/chemical reaction.

## 2. Materials and Methods

### 2.1. One-Pot/One-Step Synthesis of Radioactive Gold Nanoparticles

Auric acid and polyethylene glycol were Sigma-Aldrich^®^ Brand (Merck KGaA, Darmstadt, Germany) and ultrapure water was prepared using the Milli-Q^®^ Gradient A10 water purification system (Merck KGaA, Darmstadt, Germany). A precursor solution comprising non-radioactive gold ions (H^197^AuCl_4_, 1.0 mM), with or without the presence of polyethylene glycol (average molecular weight: 6000 Da, 1.0 mM), was subjected to synthesis in the Tsing Hua Open-pool Nuclear Reactor (THOR, National Tsing Hua University, Hsinchu, Taiwan) for 5, 15, and 30 min of irradiation. Samples were placed in the irradiation tube surrounded by cooling water, the temperature of which was maintained at 33.5 °C. The THOR simultaneously provides thermal neutrons (flux: 1.23 × 10^12^ neutrons∙cm^−2^∙s^−1^), fast neutrons (flux: 2.93 × 10^11^ neutrons∙cm^−2^∙s^−1^), and γ rays (dose rate: 1.41 kGy∙min^−1^) at an irradiation position to undertake the synthesis of RGNP. The nuclear reactor plays the role of an energy source to transform a precursor solution into an aqueous RGNP that can irradiate photons (γ rays) and β particles (Figure 1).

### 2.2. Physicochemical Characterization

Samples after irradiation were dispensed into cuvettes (SARSTEDT, Germany) with a 4 mm light path. The UV-Vis spectra of RGNP samples were measured between 300 and 840 nm using a NanoDrop 2000c spectrophotometer (Thermo Fisher Scientific, Wilmington, DE, USA). The λ_max_ and its absorbance unit (AU) were obtained. The energy spectrum of γ photons was determined using a high-purity germanium (HPGe) detector (GC1020, Canberra, Meriden, CT, USA). After irradiation in THOR for 30 min, RGNP samples were allowed to decay for at least 96 h and then subjected for 30 min of counting using HPGe detector. Aqueous RGNP samples were dripped and dried on copper grids to determine particle size using TEM (H-7650, Hitachi, Tokyo, Japan). Particle sizes of RGNP were counted using SigmaScan^®^ Pro 5 software and averaged for 200 nanoparticles.

The radioactivity of RGNP samples was determined using an automatic γ counter (2480 WIZARD^2^, PerkinElmer, Turku, Finland) at 412 keV (with a 30% energy window) for 60 s. The radioactivity of ^198^Au was presented in the unit of million counts per minute (10^6^ CPM). An aliquot of RGNP sample (0.2 mL) was filtered through an ultrafiltration device (Vivaspin 500, GE Healthcare, Uppsala, Sweden) with a 100 kDa molecular weight cut-off membrane at 5000× *g* for 10 min. The radioactivity of particulate and filtrate fractions was separately counted. The reaction yield was defined as the radioactivity ratio of the particulate fraction to the sum of filtrate and particulate fractions. Regarding preparing RGNP treatment through CED, the RGNP solutions were subjected to concentrate using ultrafiltration to achieve targeted specific radioactivity (1.0 ± 0.1 μCi/μL). The quality control approaches for measuring radioactivity and UV-Vis spectra were performed to verify physical stability before injection into animals.

### 2.3. Xenograft for Therapeutic Efficacy

Tumor-bearing mice (male NU/NU, 6–8 weeks old, BioLASCO, Taipei, Taiwan) were prepared using intracranial transplantation of IVISbrite™ U87MG-Red-Fluc cell line (PerkinElmer, Inc., Waltham, MA, USA), a light-producing cell line derived from U87 MG-human brain glioblastoma. Mice were anesthetized using isoflurane (3.5–4.0% for induction and 1.5–2.0% for maintenance) and placed in a stereotactic frame. Aliquots of cells (1 × 10^5^ cells in 3 μL of PBS) were delivered into the brain (AP: 0 mm; ML: 2 mm right; DV: 3 mm) of nude mice through a 26-gauge microneedle with 3 μL/min of flow rate. The needle was left in place post-infusion for 5 min and then was withdrawn at a rate of 1 mm/min. Four groups of mice were arranged: (1) non-treatment (sham control) group, treated with saline; (2) RGNP group, treated with RGNP (11 μ Ci in 10 μL aliquot) through CED; (3) TMZ group, treated with TMZ (5 mg/kg) through oral gavage for consecutive three days; (4) RGNP combined with TMZ group, treated as the combination of groups (2) and (3). As for the administration of RGNP, the RGNP was delivered into the brain (AP: 0 mm; ML: 2 mm right; DV: 2 mm) using a homemade CED device (Appendix A) with a 10 μL/min flow rate. All the treatments were performed on the 15th day after tumor cell implantation. Animal use protocols were reviewed and approved using the Animal Care and Use Committee of NHRI (Protocol No. NHRI-IACUC-097045-A, NHRI-IACUC-104058-A, and NHRI-IACUC-107031-A). Animals were housed in the Laboratory Animal Center at the National Health Research Institutes (Zhunan campus). The housing conditions were maintained at 24 ± 2 °C (room temperature), 50 ± 10% (relative humidity), and 12 h light/dark cycle. Animals were given ad libitum access to water and food in individually ventilated cages (IVCs). Animals have to be removed early (humane endpoint) while either 20% of body weight loss, debilitating diarrhea, labored breathing, unexpectedly moribund, cachectic, or unable to obtain food and water.

### 2.4. Bioluminescence Imaging

To evaluate the growth of U87MG-Red-Fluc cells, mice underwent in vivo bioluminescence imaging with an IVIS^®^ Spectrum (PerkinElmer Inc., Waltham, MA, USA) every 3–4 days. Mice received substrate (IVISbrite™ D-Luciferin, Cat. No. #122799, PerkinElmer Inc.) with a dose of 150 mg/kg body weight through subcutaneous injection. Mice were anesthetized with isoflurane and transferred to the heat-retaining stage for imaging. The bioluminescence of tumors was determined using 2D imaging mode (Bin:(HS)8, FOV:5, f1) at 10 min after injection of luciferin, and images were analyzed with Living Image 4.7.3 software (PerkinElmer Inc.). The radiance intensity (p/s/cm^2^/sr) presented the growth of U87MG-Red-Fluc cells.

### 2.5. ^18^F-FLT PET/CT Imaging

Mice were subjected to PET/CT imaging on the 20 days after tumor cell implantation for the non-treated group and on the 32 days after tumor cell implantation for the RGNP/TMZ-treated group. Mice fasted for 4 h before the PET/CT imaging. The animals, placed on a head holder to fix their position, were anesthetized with an isoflurane/oxygen mixture (3.5–4.0% for induction and 1.5–2.0% for maintenance). A 20 min static brain scan was acquired for 60 min after intravenous injection of 0.25 mCi of 3′-deoxy-3′-[^18^F]fluorothymidine (^18^F-FLT, Global Medical Solutions Taiwan, Ltd., Taipei, Taiwan), followed by a CT scan for anatomic co-registration. Both ^18^F-FLT PET and CT images were taken using a pre-clinical animal imaging system (FLEX Triumph^TM^; Gamma Medica-Ideas, Northridge, CA, USA). CT images were reconstructed using filtered back projection (FBP) as a matrix of 512 × 512 × 512 pixels with a pixel size of 120 μm. PET images were reconstructed through order subset expectation maximization (OSEM) as a matrix of 92 × 92 × 31 with a pixel size of 0.5 × 0.5 × 1.175 mm. The ^18^F-FLT PET images were processed using PMOD (version 3.2, PMOD Technologies Ltd., Switzerland) to depict brain tumor contour. PET images were re-sliced into a matrix of 512 × 512 × 512 with CT co-registration.

### 2.6. Histopathological Inspections

Mice were sacrificed at humane endpoints, and brain tissues were fixed using fresh 10% neutral-buffered paraformaldehyde (PFA) at room temperature for 18 h. Fixed tissues were embedded in paraffin blocks and dissected into 5 µm thick sections by the Pathology Core Laboratory at NHRI. The tissue sections were antigen retrieved, blocked, and incubated with a primary antibody targeting Ki-67 (Rabbit mAb, Cat. No. #9027, 1:300 dilution; Cell Signaling Technology^®^, Danvers, MA, USA) or proliferative cell nuclear antigens (PCNA) (Rabbit mAb, Cat. No. #13110, 1:8000 dilution; Cell Signaling Technology^®^) at 4 °C overnight and subsequently incubated with a secondary antibody (SignalStain^®^ Boost IHC Detection Reagent, HRP Rabbit, Cat. No. #8114, Cell Signaling Technology^®^) at room temperature for 2 h. Slides were stained with DAB for immunohistochemistry (IHC) inspection, in which IHC for Ki-67 and PCNA was performed on different tissue dissections.

Further, another tissue section was stained with hematoxylin and eosin (H&E) to check cellular and tissue features. Hematoxylin stained the nuclear components with a purplish blue color and eosin stained the cytoplasmic components a pink color. Images were taken using a microscope (Leica DM2500, Leica Microsystems, Wetzlar, Germany) at 100× and 200× magnification. For tissues containing RGNP, a Geiger–Müller counter was employed to survey the radioactivity decaying to the background level before subjecting tissue dissections to the abovementioned procedures.

### 2.7. Biodistribution of Radioactive Gold Nanoparticles

To investigate whether post-CED RGNP could be re-distributed into blood circulation, RGNP was administered to tumor-bearing mice (n = 7) through CED at 15 days post-implantation of U87MG-Red F-luc cells. At 24 h and 72 h after CED, the blood samples (10 μL) were collected to determine RGNP. Radioactivity measurements for ^198^Au in RGNP in blood were performed using an automatic γ counter (2480 WIZARD^2^, PerkinElmer, Turku, Finland) and the percentage of the injected dose of RGNP in 1.0 mL blood (%ID/mL) was recorded. To explore the biodistribution of RGNP in different organs, tumor-bearing mice (n = 8) were administered with RGNP alone or a combination of RGNP plus TMZ, and were then euthanized at a humane endpoint (26–57 days after tumor cell implantation) to collect organs, including brain, heart, lung, liver, spleen, pancreas, kidney, stomach, intestine, and carcass. Radioactivity measurements for ^198^Au in RGNP in different organs were performed using an automatic γ counter and the percentage of injected dose per gram sample wet weight (%ID/g) was recorded.

## 3. Results and Discussion

### 3.1. One-Pot/One-Step Reactions in the Nuclear Reactor

Two recipes of precursor solution were subjected to irradiation in the THOR for the preparation of RGNP. Figure 1 shows the precursor solution converted to pink or burgundy after irradiation. UV-Vis spectra are crucial in providing characteristic information on the consumption of Au(III) ions and the formation of gold nanoparticles (GNP). For the precursor solution containing HAuCl_4_ alone (Figure 1A), the peak at 310 nm, referring to trivalent Au(III) ions, slightly drops after 5 min followed by sudden disappearance for 15 min of irradiation, implying the consumption of HAuCl_4_ [49]. The occurrence of a broadband signal around 568–580 nm (Table 1) should indicate GNPs with sizes larger than 100 nm, resulting from aggregation or agglomeration [50,51]. In terms of the mixture containing HAuCl_4_ and PEG6000, the peak at 310 nm vanishes and is accompanied by a dramatic rise of a classical surface plasmon band (SPB) at 520 nm for only 5 min of irradiation (Figure 1B), suggesting Au(III) ions were rapidly exhausted and converted into GNP. The intensities of the SPB persist for the reaction from 5 min to 30 min (Table 1), indicating completeness of GNP formation compared to the poor yield of GNP synthesized by HAuCl_4_ alone. Radioactive gold nanoparticles (RGNPs) were successfully synthesized without using radioactive precursors and reducing agents.

The radioactivity of samples was shown in a million counts per minute for the 412 keV γ photons (Table 1). Levels of radioactivity were tunable by adjusting irradiation time in the nuclear reactor, demonstrating a more than 5-fold increase in radioactivity while prolonging the irradiation time from 5 to 30 min. Through ultrafiltration, the yield of RGNP was determined by dividing the radioactivity of particulate fraction by the total radioactivity of particulate and filtrate fractions. Table 1 shows that synthesis yields of RGNP using HAuCl_4_ precursor alone vary from 53.6% through 61.5% to 100% by changing irradiation time from 5 through 15 to 30 min. Regarding precursor solutions consisting of HAuCl_4_ and PEG6000, yields of RGNP were maintained at 100% from 5 through 15 to 30 min of reaction. The completeness of producing RGNP shows perfect accordance with the prediction conducted using SPB measurements (Figure 1B).

### 3.2. Determination of Particle Sizes

To determine the particle size of RGNP, both UV-Vis and transmission electron microscopy (TEM) provide complementary information. UV-Vis spectra are feasible in verifying the completeness of the reaction. They are also convenient for predicting the sizes of RGNP before the radioactivity decays to an acceptable level for TEM analysis. Link et al. observed that characteristic SPBs at 517, 520, 521, 533, and 575 nm correspond to GNP with average sizes of 9, 15, 22, 48, and 99 nm [50,51]. Haiss et al. reported empirical equations to predict the sizes of GNP based on the SPBs in UV-Vis spectra [52]. Using the methods mentioned above, the RGNP sizes shown in Figure 1A and Figure 1B have been estimated to be larger than 99 nm and near 15 nm, respectively. The SPB located at 520 nm (Figure 1B) demonstrates a coincidence of RGNP prepared in this study and classical 15 nm GNPs introduced by Turkevich [53] and then elaborated by Frens [54].

TEM is the gold standard to validate the prediction by characteristic SPBs in UV-Vis spectra. For a precursor solution containing HAuCl_4_ alone, two distinct populations of RGNP were observed at 8.6 nm and 105.9 nm for only 5 min of reaction (Table 1). The RGNP grew up to ~140 nm while we extended the reaction time to 15 min and longer. Regarding the precursor solution containing HAuCl_4_ and PEG6000, the reaction finished at 5 min to obtain RGNP with an average size of 14.6 ± 3.7 nm. The particle size was 15.9 ± 3.7 nm while the reaction time was extended to 30 min (Figure 2A), indicating highly favorable agreement with ~15 nm of the size predicted using UV-Vis spectra. The PEG6000 plays the role of stabilizer, such as Tween 20 suggested by Aslan et al. [55], to protect gold nanoparticles from aggregation.

### 3.3. Harnessing Nuclear Energy to Generate Radioactive Gold Nanoparticles

The particulate fraction collected on the ultrafiltration membrane was subjected to high-purity germanium (HPGe) detector to inspect radionuclidic purity. Figure 2B presents the relative abundance of γ photons with a prominent peak at 412.2 keV (100%) and minor peaks at 674.5 keV (0.52%) and 1083.2 keV (0.09%). All of these peaks agree with the decay scheme of ^198^Au (half-life: 2.69 days), presenting γ photons peaked at 411.8, 675.9, and 1087.7 keV [41]. The ^198^Au is generated from natural gold (^197^Au) through a neutron capture reaction. The formation of RGNP containing ^198^Au should attribute to the radiation-induced reactions at the exact moment of the irradiation procedure [56]. Katti et al. have reported the preparation of radioactive auric acid (H^198^AuCl_4_) through neutron irradiation of pure gold foil (^197^Au) followed by dissolving the irradiated gold foil into aqua regia [57]. They then employed H^198^AuCl_4_ as a precursor to synthesize RGNP using the conventional method; however, this procedure was tedious, and extensive radiation safety concerns were notified from starting materials, products, and waste. We have developed two approaches to synthesizing RGNP without using radioactive precursors (Appendix A) [39]. The one-pot/one-step reaction and the three-step reaction demonstrated the same radiochemical property along with similar physical properties despite that reaction types and surface ligands were totally different (Appendix A).

### 3.4. Proposed Mechanism

Neutrons and γ rays in nuclear reactors contribute to radiation-induced reactions that are not easily reproduced in a typical environment [58,59]. The nuclear reactor is a concomitant radiation field to provide thermal neutrons (n_th_), fast neutrons (n_f_), and γ photons (γ) to enable a series of reactions [60]. Thermal neutrons react with ^197^Au nuclei through a neutron capture reaction (Equation (1)) that converts ^197^Au atoms to radioactive ^198^Au atoms.
^197^Au + n_th_ → ^198^Au + γ (1)

Fast neutrons possessing nearly an equivalent mass of protons can interact efficiently with hydrogen atoms of water molecules through elastic/inelastic collision. The billiard ball-like collision between fast neutrons and hydrogen atoms can generate ejected protons (H^†^) shown in Equation (2). The energy distribution of ejected protons varies from zero to a maximum in a head-on collision, which is ultimately transferred to other molecules through ionization or excitation in the energy dissipation processes.
H (in H_2_O) + n_f_ → H^+^ (ejected protons)(2)

Previous studies have indicated that ionizing radiations, including γ-rays, x-rays, electrons, and charged particles, lead to the radiolysis of water and produce various radiolytic species, such as hydrated electrons (e^−^_aq_) and hydrogen radicals (H·), to facilitate a reductive/oxidative reaction in aqueous environments [40,60,61,62]. Ejected protons and γ-photons presented in a nuclear reactor can thus interact with water molecules in the energy dissipation process to derive primary radiolysis species shown in Equation (3). All of the above reactions occur in a time scale of 10^−13^ s or less.
(3)$$H_{2}O\overset{ejected\;protons\;\&\;\gamma -photons}{\to }H^{+},\;e_{aq}^{-},\;H,\;HO,\;{HO}_{2},\;H_{2}O_{2},\;H_{2}$$

In an aqueous environment, incident ionizing radiations, such as ejected protons and γ-photons, generate “spurs” along their moving tracks in which many primary radiolysis species are generated. These primary species of radiolysis further induce a series of subsequent reactions. As shown in Equations (4) and (5), both e^−^_aq_ and H· are extremely high-potent reducing agents and feasible to turn trivalent gold (Au^III^) into zero-valence gold (Au^0^). These reactions occur at a time scale of 10^−11^ s, during which thermodynamic equilibrium can be established.
^198^Au^III^/^197^Au^III^ + H· → ^198^Au^0^/^197^Au^0^ + H^†^(4)
^198^Au^III^/^197^Au^III^ + e^−^_aq_ → ^198^Au^0^/^197^Au^0^(5)

Subsequently, a chemical equilibrium stage occurs on a 10^−8^ s or longer time scale. The ^198^Au-RGNP is produced through a coalescing process of ^197^Au together with ^198^Au atoms, as shown in Equation (6). Extending irradiation time for precursor solution in a nuclear reactor can convert more ^197^Au into ^198^Au through Equation (1), which can raise radioactivity for RGNP.
x ^198^Au^0^ + y^197^Au^0^ → RGNP(6)

### 3.5. Therapeutic Efficacy

Survival curves are employed to demonstrate the therapeutic efficacy of different treatments compared to the non-treatment group. Figure 3A illustrates the survival percentages by the days after tumor cell implantation for groups treated using RGNP, TMZ, and the combination of RGNP/TMZ compared to the non-treated group. The maximal survival for RGNP/TMZ treatment group was 176 days; however, that for the non-treated group was less than 35 days. The mean survival was 28.62 ± 0.77, 38.44 ± 2.20, 42.85 ± 2.50, and 68.88 ± 9.65 days for non-treated, RGNP(CED), TMZ(oral), and RGNP/TMZ groups, respectively, in which therapeutic efficacy of treatment groups was better than the non-treated group with significant difference (all *p-*values: <0.0001) (Figure 3B). The median survival was 28, 35, 44, and 54 days (Table 2), showing the combination of RGNP and TMZ treatment superior to RGNP alone or TMZ alone treatments. Furthermore, there was no significant difference (*p-*value: 0.158) between the mean survival of the RGNP group and that of the TMZ group (Table 2).

Figure 4 demonstrates bioluminescence images of glioblastoma-bearing mice acquired using IVIS, showing very rapid growth of tumors of the non-treatment group within 32 days after intracranial injection of tumor cells. Similarly, glioblastoma-bearing mice treated using TMZ alone showed fast tumor growth within 46 days. The tumor sizes of the RGNP/TMZ and RGNP alone groups grew slowly within 39 days. In contrast, the combination treatment of RGNP/TMZ sustained inhibiting tumor growth until 49 days compared to the standalone treatment of RGNP. The RGNP group presented slight progress of tumors 46–49 days after intracranial injection of tumor cells. Both mean survival days (Table 2) and bioluminescence images (Figure 4) give us insight into the synergistic effect attributed to the combined therapy of RGNP infused through CED with oral administration of TMZ.

To verify the locoregional therapeutic efficacy, Ki-67 and proliferating cell nuclear antigen (PCNA) were essential biomarkers to determine proliferative activity, tumor grade, and malignancy [63]. Figure 5 illustrates highly concentrated expression of Ki-67 in tumor regions of the non-treatment group and mice treated with TMZ; nevertheless, a little scattered expression of Ki-67 was observed in the nearby regions of glioblastoma treated with either RGNP alone or a combination of RGNP/TMZ. Remarkably, the PCNA was highly suppressed around RGNP that infused through CED no matter in RGNP alone or RGNP/TMZ group. The TMZ might be a radiosensitizer to boost localized radiotherapy against glioblastoma since previous work indicated TMZ leads to cell cycle arrest at the G2/M phase [64]. In terms of the anti-PCNA image of the RGNP/TMZ group in Figure 5, there is a significant portion of tumor cells strongly expressing PCNA, presenting a more heterogenous expression of PCNA than Ki-67 in a microscopic view in contrast to macroscopic views of survival (Table 2) and tumor size (Figure 4). Kayaselcuk et al. have indicated that both PCNA and Ki-67 are invaluable nuclear markers to determine the grade and proliferative status of central nervous system (CNS) tumors; however, Ki-67 is a more specific marker to present the proliferative index [63].

Further, the ^18^F-FLT PET/CT imaging was employed to evaluate the efficacy of RGNP compared to TMZ treatment (Figure 6B,E), showing RGNP superior to TMZ for inhibiting tumor growth. Confirmed inhibition of PCNA nearby the brain tumor was observed in the RGNP treatment group (Figure 6A,C) instead of a gentle expression of PCNA nearby the brain tumor in the TMZ treatment group (Figure 6D,F), which was in good agreement with bioluminescence images performed using IVIS (Figure 4).

### 3.6. Biodistribution of RGNP after Convection-Enhanced Delivery

Blood samples were collected after delivering RGNP into the brain of glioblastoma-bearing mice using CED to explore whether RGNP could migrate across the blood–brain barrier. Neither at 24h nor 72h after CED, we can not observe RGNP in blood by detecting 412 keV γ photons in an automatic γ counter, which suggests RGNP can be dominantly accumulated in brains but not in blood circulation. We, therefore, indicate that retention of RGNP in the brain limits the re-distribution of RGNP to other organs. Regarding relative long-term evaluations, mice who fulfilled the early removal criteria (humane endpoint at 26–57 days post-implantation of tumor cells) were euthanized to collect organs for determining the radioactivity of RGNP. The mean value of biodistribution for RGNP in the brain was 91.71 ± 4.51% (Figure 7A). The RGNP used for treating glioblastoma-bearing mice contains PEG6000. We propose that PEG6000 molecules tentatively protect the surface of RGNP to avoid aggregation. After being delivered into the brain, RGNP would aggregate while interacting with proteins in the tumoral/peritumoral regions leading to protein corona on RGNP. Therefore, we propose that RGNP does not pass through the blood–brain barrier (BBB) because of in situ aggregation triggered post-administration.

In contrast, the presence of RGNP in carcass was merely 4.37 ± 3.82%, and those in other organs were less than 0.5%. To demonstrate biodistribution in another way (concentration-wise), the RGNP in the brain was 140.39 ± 22.84%ID/g (Figure 7B), presenting the highest concentration of RGNP in the brain compared to that in other organs. The RGNP in the spleen, liver, and carcass were 4.03, 0.62, and 0.80%ID/g, respectively, which showed that RGNP was effectively infused into the brain through CED and minimized its re-distribution into circulation as well as other organs. Compared to previous work, RGNP was less than 0.3% in the brain, while RGNP was delivered through intravenous injection [39]. The biodistribution of different types of RGNPs in the liver of healthy ICR mice varied from 12.1% to 88.4% while they were delivered through intravenous administration. RGNP in the spleen varied from 0.2% to 3.8%, whereas that in the carcass varied from 3.8% to 65.4%. Previous studies have indicated that drugs infused through CED can provide an intraparenchymal concentration of drugs 1000-fold more extraordinary than those through intravenous administration [44,45,46,47]. The RGNP showed an average of 140.39% ID/g in the brain, which was 7- to 10-fold more prominent than the epirubicin-loaded nanoparticle (14.10–20.89%ID/g in the brain) described in a previous study [47]. Therefore, we suggest CED should be invaluable to infusing RGNP for brain tumor therapy.

## 4. Conclusions

Neutrons and γ photons in a nuclear reactor enable chemical/nuclear reactions to synthesize gold nanoparticles and simultaneously render them radioactivity for locoregional radiotherapy. In this work, a one-pot/one-step approach harnesses nuclear energy to generate RGNP without radioactive precursors and reducing agents. The RGNP comprises ^198^Au atoms that can irradiate β particles and γ photons, the former facilitates killing tumor cells, and the latter is employed to investigate the biodisposition of RGNP. The RGNP, infused through convection-enhanced delivery combined with TMZ, provided through oral administration, takes advantage of synergistic effects to treat glioblastoma xenografts. Bioluminescence images, positron emission tomography, and immunohistochemistry inspection all verify the survival curves, indicating superior therapeutic efficacy attributed to the combination of RGNP and TMZ.

## Data Availability

Not applicable.

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
