# Peer review of "Harnessing Nuclear Energy to Gold Nanoparticles for the Concurrent Chemoradiotherapy of Glioblastoma"

_nanomaterials, 2023, doi:10.3390/nano13212821_

Round 1

Reviewer 1 Report

Here Li et al demonstrated a one-step process to synthesize RGNP (198Au) and showed its therapeutic potency towards Glioblastoma (GBM)-xenograft model following CED & TMZ co-treatment.

The authors showed significantly longer survival period of the mice, attributed to tumor growth retardation. The authors further evaluated distribution of RGNP to other organs, showing the greatly localized delivery mediated by CED. 

While the authors present significant effort in this article, there are several aspects of the study that warrant careful consideration before its acceptance:

- Can the authors present results regarding the physical stability of RGNP? Related to this, how is the storage condition of the RGNP?

- Localized intracranial injection by CED ensures maximum efficacy of proposed RGNP while minimizing systemic distribution (as the authors show). However, this is hard to be translated into clinical settings. Are there toxicity concerns of the RGNP when delivered systemically?

- Related to above, can the authors explain possible reasons why RGNPs do not pass through BBB, despite its small size & solid nature?

- In fig 5, for RGNP+TMZ anti-PCNA image, there are significant portion of tumor strongly expressing PCNA, indicating active proliferation. Can the authors address this, as it seems to contradict fig 4 result in which tumor growth is almost completely suppressed?

Reviewer 2 Report

The Authors have submitted a manuscript regarding the synthesis and preliminary in vivo evaluation of radioactive gold nanoparticles.

The manuscript is interesting, and the presentation logical. It addresses a topic of relevance for both materials science and nanomedicine, and suits with the Journal scope. Overall, the manuscript can be accepted for publication after the following minors will be addressed:

- “Chemotherapeutics-conjugated GNP targeting to tumors is attributed to the enhanced permeation and retention (EPR) effect, leading to more internalized drugs in tumor cells than molecular drugs”. The Authors should underline that this effect happens only for some type of carcinoma (doi: 10.1038/s41467-018-03705-y)

- “Just three of them are GNP-based drug carriers for anticancer because GNP's restricted drug loading content weakens its clinical applications”. This sentence is not completely true. Indeed, the drug-loading is one of the concerns while the most important for clinical translation is the persistence of gold (doi: 10.1002/ppsc.201800464 and 10.1039/d0na00521e and associated). The Authors should discuss this point in the introduction and in the discussion (Biodistribution of RGNP section) as their AuNPs are >5nm.

- Authors should better discuss the chosen target (Glioblastoma multiforme). AuNPs are especially (or it is expected to be) efficient for this kind of cancer? Or are there other reasons?

- the choice of convection-enhanced delivery reduces the overall impact of the manuscript, as one of the main concerns for the treatment of GBM is passing the BBB (doi: 10.1016/j.nbd.2009.07.030). The Authors should better discuss this point.    

Reviewer 3 Report

It is better to write nuclide in superscript in Abstract and Keywords.

Line 25: 197Au  →  197Au

Line 26, 36: 198Au  →  198Au

In Introduction, 

Line40: Faraday M's paper is only No.3. Isn't it strange to write reference as [1-3]?

No.1 explains Faraday's achievements, so I feel that the place to quote is a little different.

Wouldn't it be better to put [1, 2] after (GNP)?

Line 22, 89, 90, 113, 115, 126, 132, 211, 218, 279, 280, 287, 301, 305, 424, 453, 457: 

gamma   →  γ   

Line 89, 92, 115, 457, : beta   →  β

It is better to unify the notation of min and minute (Line 135, 176, 177, 238, 278, 280). 

It is better to unify the notation of h and hour (Line 128, 174, 189, 196, 279). 

It is better to unify the notation of sec and second (Line 134, 329, 338, 344).

The multiplication sign was mixed with x. Better to unify (for example Line 183, 184 and 186).

Line 169: cm2  →  cm2

Line 171: I don't quite understand the meaning of the title.

Line 202: Missing company description.

Please indicate that you have obtained permission for animal study (e.g. whether or not permission has been obtained from your institution's Animal Care and Use Committee, permission number, etc.). Also indicate the maintenance conditions of the animals. Since mice were maintained for a long time, it is sometimes asked whether the animals are properly handled.

Round 2

Reviewer 1 Report

The authors have provided sufficient revision and comments regarding my input. 

I would suggest the acceptance of this article.